# The Role of Pre-Clinical 3-Dimensional Models of Osteosarcoma

**DOI:** 10.3390/ijms21155499

**Published:** 2020-07-31

**Authors:** Hannah L. Smith, Stephen A. Beers, Juliet C. Gray, Janos M. Kanczler

**Affiliations:** 1Antibody and Vaccine Group, Centre for Cancer Immunology, University of Southampton, Faculty of Medicine, Southampton General Hospital, Tremona Road, Southampton SO16 6YD, UK; hannah.smith@soton.ac.uk (H.L.S.); s.a.beers@soton.ac.uk (S.A.B.); j.c.gray@soton.ac.uk (J.C.G.); 2Bone and Joint Group, Institute for Developmental Sciences, University of Southampton, Faculty of Medicine, Southampton General Hospital, Tremona Road, Southampton SO16 6YD, UK

**Keywords:** osteosarcoma, 3D model, in vivo, ex vivo, in ovo

## Abstract

Treatment for osteosarcoma (OS) has been largely unchanged for several decades, with typical therapies being a mixture of chemotherapy and surgery. Although therapeutic targets and products against cancer are being continually developed, only a limited number have proved therapeutically active in OS. Thus, the understanding of the OS microenvironment and its interactions are becoming more important in developing new therapies. Three-dimensional (3D) models are important tools in increasing our understanding of complex mechanisms and interactions, such as in OS. In this review, in vivo animal models, in vitro 3D models and in ovo chorioallantoic membrane (CAM) models, are evaluated and discussed as to their contribution in understanding the progressive nature of OS, and cancer research. We aim to provide insight and prospective future directions into the potential translation of 3D models in OS.

## 1. Introduction

Osteosarcoma (OS), although a rare type of cancer, is the most frequent primary bone tumour, and accounts for over 10% of all solid tumours in adolescents [1]. The incidence of OS is bimodal, with peaks in adolescence and in the elderly [2]. Although the aetiology of OS is still unknown, evidence suggests that it is a genomically unstable disease with abnormal karyotypes [3,4]; a relatively high percentage of patients have pre-disposing somatic P53 and Retinoblastoma (Rb) deletions/point mutations [5,6,7,8]. In younger patients, OS most commonly invades the metaphyses of long bones, including the proximal tibia, humerus and distal femur [9]. These tumours have the ability to produce metastases that translocate and form most frequently in the lungs [10]. Five year event free survival rates for OS are reported to be approximately 50%, but are much lower in patients with metastatic disease, who have a 5 year event free survival rate of 30% [11]. The second peak of incidence of OS is found in patients aged 60–85 years, where it is generally considered to be a secondary neoplasm. In these patients, OS occurs more commonly in axial locations, where the bone has underlying abnormalities or has been previously irradiated [9]. The 5-year survival rate in this population is reported to be 24.2% [9].

The majority of patients with OS undergo a combination of neoadjuvant chemotherapy followed by surgical resection of the tumour. The first line chemotherapy used to treat OS has been largely unchanged for several decades, with the majority of patients receiving regimens consisting of methotrexate, doxorubicin and cisplatin. In 2001, Mifamurtide, an immunomodulating muramyl tripeptide, was approved by the US Food and Drug Administration (FDA) [12] and is now used in routine clinical practice as a component of front line therapy, although the impact on long term survival is unclear. Several novel therapeutic agents are undergoing clinical trials, including bisphosphonates and inhaled granulocyte macrophage colony stimulating factor (GM-CSF). Bisphosphonates are routinely used to treat osteoporosis, but they have been found to promote anti-tumour immunity in vitro, by inhibiting proliferation and inducing apoptosis of cancer cells [13]. GM-CSF is a cytokine secreted by leukocytes that stimulates proliferation of multipotent progenitor cells. The aim of the inhaled therapy is to induce the expression of Fas and Fas ligand in the metastatic form of OS, which is downregulated on OS metastatic cells [13]. Therapeutic targets and products are being continually discovered and developed to try to promote regression and cell death, but not many are therapeutically active in OS. Therefore, there is an increased need to understand the growth and function of OS alongside its microenvironment, in order to produce more robust and active therapies.

## 2. Biological Understanding of OS

The cell of origin for OS is unclear, with current evidence suggesting it occurs somewhere on the mesenchymal stem cell (MSC) to osteoblast differentiation pathway [14]. Unfortunately, key phenotypic markers to accurately define MSCs have not yet been found, but they are understood to be stem-cell like precursors of important structural cells in bone, including osteoblasts, adipocytes and chondrocytes [15]. MSCs are proposed to be one of the cell types found in bone marrow stromal cells (BMSCs), which are progenitors of skeletal components and have been found to inhibit anti-tumour immune responses [16], as well as promote tumour growth and metastasis [17,18]. OS has been graded into seven subtypes: osteoblastic, fibroblastic, chondroblastic, epithelioid, giant-cell rich, small cell and telangiectatic [14,19], but it is unclear if these subtypes are associated with distinct genetic mutations. Bone formation and bone resorption are closely regulated by osteoblasts and osteoclasts, respectively, but their function can be altered in the tumour microenvironment. Osteoclasts are regulated by the secretion of receptor activator of nuclear factor–κB ligand (RANKL) by osteoblasts. RANKL and other osteoclast stimulating factors, including Interleukin-6 (IL-6) and Interleukin-11 (IL-11), are also secreted by tumour cells in bone metastases [20] and OS [21,22], leading to an increase in bone resorption. This resorption releases growth factors, including transforming growth factor beta (TGFβ) and bone morphogenetic protein (BMP) that interact with the microenvironment to stimulate tumour growth, which in turn stimulates the tumour to induce further bone resorption [23].

The tumour microenvironment of OS is complex and variable between patients. The interactions between cell–cell and cell–extracellular matrix (ECM) are still not completely understood and are being recognised as key factors in developing future targets against this disease. Bone marrow is an important part of the immune system, with lymphocytes consisting of between 8% and 20% of all bone marrow mononuclear cells [24]. Lymphocytes and other immune cells are transported through the bone marrow (BM) via blood vessels, and are found throughout the stroma and in unique follicle-like structures of lymphocytes [24], see Figure 1. The interaction of these immune cells with structural bone cells, like osteoblasts and stromal cells, while in the presence of OS is poorly understood. It has been demonstrated that while cancer cells can recruit immune suppressive cells, including regulatory T cells (Tregs) and tumour associated macrophages (TAMs), to the tumour site, they can also manipulate the function of immune cells into a tumour-promoting phenotype, facilitating tumour growth [25]. These immune cells interact with other cells with an altered phenotype in the BM including BMSCs.

## 3. Pre-Clinical Modelling of OS-3D Models

3D models offer the potential to better understand microenvironmental interactions compared to 2-dimensional (2D) in vitro assays. Whilst the latter have been essential in the field of cancer research; producing novel insights into characterising protein production, cell biology and the tissue morphology of various subsets of cancer [26], they have many limitations. These include inducing changes in polarity and morphology as well as causing disruption of cellular and extracellular interactions [26]. These limitations have led to an increase in the number of research groups developing assays to better imitate in vivo conditions by using 3D models. There are a growing number of 3D models that have been developed, and in this review, we will explore the benefits and disadvantages of in vivo, in vitro and in ovo 3D models of OS.

## 4. In Vivo Animal Models

As the origin and pathogenesis of OS is still unknown, establishing a robust and representative animal model, that mimics all aspects of the human disease, is challenging. Initial rodent models for OS were developed by exposing the tibia to chemical and radioactive carcinogens, which produced histologically accurate tumours and can still be utilised to represent the DNA damage effect on pathogenesis [27]. Despite the success in generating this model, it does not represent the sporadic and spontaneous generation of this disease in humans and is not routinely used to represent the aetiology of OS [10]. The establishment of immunocompromised mice allowed for the inoculation of a variety of human OS cell lines; these can be injected subcutaneously, although the inoculation of OS cell lines or cell grafts directly into the femur are considered more therapeutically relevant as subcutaneous injection only allows for ectopic bone formation. These models have had a range of success in establishing OS migration, especially as the murine model allows for metastasis [28]. Moreover, these models are a useful tool for screening drugs [29,30]. Arguably, the induction of OS by intraosseous injection is one of the most relevant murine models for studying this disease, with studies recapitulating OS tumour heterogeneity [31], as well as providing biological understanding of activation pathways [32]. The fundamental limitation of this model is the use of mature OS cells, which does not provide information on tumour initiation. The lack of a tumour microenvironment before implantation, and a lack of an intact immune system, may also mean key interactions are lost [33].

The introduction of gene targeting technologies allowed for the development of transgenic mice. There are several genetically modified mouse models for OS; most are based on the P53 and Rb mutations [10]. When mice were generated with mutations in the Rb gene alone, it was found that the Rb gene is essential for normal mouse development, with some strains producing serious defects, making breeding difficult [34]. Moreover, even when such mice have been established, they have not been found to develop OS [35]. There has been more success in P53 transgenic mice, where spontaneous OS development was observed [36]. This was accelerated with the combination of p53 and Rb mutations [35,37]. Importantly mice with P53 mutations also developed lymphomas, carcinomas and adenocarcinomas [36,38], sometimes reaching a humane endpoint before the establishment or characterisation of OS could occur. While mouse models allow us to gain some insight into the development of OS, the high cost and limited representation of human OS, means effective and robust alternatives are needed.

Canine OS disease is more common than the human equivalent, but it has been shown to be very similar histologically and have parallel genetic features to human OS. The primary treatment method for human OS of surgery and chemotherapy is also the main treatment used in canine cases, which have been found to show a similar level of response [39]. The level of similarity between the two species shows an increased level of relevancy compared to other animal models, with studies into canine OS gene expression after chemotherapy identifying significantly differentiated pathways between non-responders and responders, which were consequently found to be mirrored in human OS [40]. Although gene expression pathways in canine OS are not always found in human OS [41], the frequency of canine OS compared to the sporadic rare nature of human OS, as well as the closer histological similarities than mouse models, allows an increased opportunity to identify further biological and genetic pathways that can impact new therapies and treatments.

## 5. In Vitro 3D Models

As mouse models do have limitations, especially regarding early development of OS, many researchers are reducing the level of animal work by developing in vitro 3D models. These models can be broadly separated into two groups, those with a scaffold and those without. 3D models without a scaffold normally entail the formation of spheroids or tumour-like structures. One method of looking at spheroid formation of OS is using liquid overlays or ultra-low binding plates [42]. The liquid overlay, which can be a type of agar/agarose [43] or poly-hydroxyethyl methacrylate (HEMA) [44], is used to coat the plate, preventing the cells from adhering to the surface of the vessel enhancing cellular interactions allowing them to form spheroids. These methods have been used to study how hypoxic changes [45] can increase spheroid adhesion, and how drug responses can differ between spheroids and 2D cultures [44], in both OS cell lines and other cancer cell lines. Ultra-low binding plates have been used more extensively to compare differences in chemoresistance of spheroids and monolayer cultures [46] and have been used to identify miR-335 as a potential regulator of cancer stem cell (CSC) activity [47]. An alternative scaffold free method is the hanging drop method, this technique utilises gravity to generate multi-cell co-cultures [42], and have been combined and compared with 2D cultures [48] to show that cells grown in the hanging drop technique express increased levels of VEGF, a precursor to angiogenesis, and other bone proteins compared to 2D cultures [48]. While these scaffold free techniques are relatively low cost and have the advantage of being high throughput, they lack a number of components of the tumour microenvironment, including the ECM–cell interactions, which influences the formation and growth of OS.

In the bone microenvironment mechanical signals are very important in modulating tumour behaviour [49]. Although the generation of scaffold free spheroid cultures are key in OS development, the use of scaffolds can be more clinically relevant, as they allow for cell adhesion and an added dimension of cell-ECM signalling compared to scaffold free models. The use of silk sponges has already indicated that there is an increase in chemoresistance when OS cells are cultured in 3D compared to 2D models [50], although this material is not naturally found in the tumour microenvironment which could affect the relevance of this scaffold. The material selected to form a scaffold must be chosen appropriately to allow for correct functional environmental signalling. Scaffolds can be made from natural material (collagen, Matrigel, Silk), or synthetic (Hydrogel, polyethylene glycol, ceramics) [42,49]. The hard brittle surface and mechanical stiffness of ceramic scaffolds exhibit biocompatibility to bone because of their structural similarity [51], and studies have shown osteoblast differentiation and proliferation are enhanced when interacting with ceramic [52], unfortunately the brittleness of the scaffold means it has limited applications [53]. The success of synthetic scaffolds is the flexibility in fabrication and architecture whilst being able to control degradation [54], but in turn they can have reduced levels of bioactivity [51]. Natural scaffolds are almost the opposite to synthetic scaffolds in that they generally have good levels of bioactivity but can show increased levels of degradation [49]. The choice of scaffold will depend on the scientific question, whether mechanical stiffness or slow degradation is required. A mixture of synthetic and natural scaffolds have been achieved in stem cell research [55] that can overcome some of these disadvantages, but there is a limited amount of published work showing their use with OS, especially when co-culturing with ECM and other important cell types found in the microenvironment.

ECM is a major component of the tumour microenvironment, with studies showing that it acts in a pro-tumorigenic capacity, enhancing OS cell invasion, with some evidence suggesting it provides a protective niche for drug resistance [56]. This has been studied mainly in 2D in vitro assays using multiple components of ECM, including fibroblasts [57] and acellular proteins [56]. There have also been few 3D models incorporating components of the ECM, including the use of collagen sponges [49] and growth factors [58] to simulate aspects of the tumour microenvironment. The complete and integrated understanding of the OS ECM interaction is important as it makes up a large part of the vascular network, and one of the main limitations of in vitro 3D models is the lack of angiogenesis, important in tumour growth. The ability to develop angiogenesis is possible in both in vivo and in ovo experiments and is a key aspect for all models while trying to elucidate the origins of OS.

## 6. In Ovo Chorioallantoic Membrane Models

The chorioallantoic membrane (CAM) is a dense vascular network that rapidly develops in a fertilised chicken egg, it is formed 4-5 days after fertilisation by the fusion of the chorion and the allantoic vesicle [59]. Its main role is as a respiratory organ for the embryo, storing waste products and absorbing calcium from the shell [59], see Figure 2. The use of the CAM to investigate tumour growth started back in the early 1900s, where Murphy (1913) grew rat sarcoma tumours in CAM models and transplanted them into adult chickens. Here he noted the tumour did not adapt to include avian cells, causing the tumour to be rejected from the adult chickens [60]. This natural immunodeficiency of the embryo is a unique property of the CAM assay, which allows for in depth phenotypic analysis on the implanted tissue of interest, without the host avian cells rejecting or majorly altering the tissue [61]. One of the most important benefits of the CAM assay over 2D methods is the ability to study angiogenesis. Angiogenesis is important in tumour development [62], thus precise monitoring with and without anti-angiogenic agents could lead to new targets for treating OS [59]. Angiogenesis is also important in trying to accurately and robustly replicate the OS microenvironment to characterise complete cellular interactions and signalling methods. Although OS has not been widely studied in this model, there are many other cancer cell lines that have been studied in CAM assays, with and without a support system for implantation, which can include collagen [63] and Matrigel [64] grafts, plastic rings [65], and sponges [66]. These implantation techniques have been used to look at invasion, metastasis, and drug development [63,64,65,66].

One of the few OS experiments involving the CAM model was by Balke et al. (2010) where they implanted eight different OS cell lines into the CAM model to compare how the cell line affects tumour development and formation. Of the eight cell lines only three (MNNG-HOS, U2OS and Saos-2) formed solid, vascular tumours which were more than 2 mm in size [59]. Unfortunately, none of the developed tumours showed osteoid formation that is normally found in human disease, which could have resulted from a lack of mature bone cells or growth factors in the model. The low cost and ease of purchase allows for a high yield and quick turnover of CAM data, but there is only a short viable timespan of a CAM assay (up to 18 days, 2/3rd gestation). This timespan does not allow for long term studies, which could be key for OS establishment. This can be partly negated by transferring the implant over multiple CAM models [60,67], allowing for a longer growth period, but we do not know what impact this transfer has on the establishment of the tumours.

## 7. Conclusions

Outcome for OS is largely unchanged over several decades and novel therapeutic drug strategies are needed. With more awareness on the heterogeneity and complexity of the tumour microenvironment, robust and informative 3D models are being developed to mimic the cellular interactions found in human OS. This review summarises the current forms of 3D models used to study OS, in vivo, in vitro and in ovo. In theory, in vivo mouse studies are advantageous models to investigate OS development and drug efficacy [29,36]. Mice have a close genetic lineage to humans, and non-immunocompromised mice have a complete immune system. Murine models for OS have given further insights into methods of tumour invasion and metastasis, when compared to current in vitro and in ovo 3D models [28]. Unfortunately, intraosseous injection of OS cells into the femur is a highly skilled and very costly process, which does not give insight to the priming of the microenvironment before tumour initiation. Alternative subcutaneous implantation and grafts also do not answer the questions on OS establishment, and do not replicate the human microenvironment as they lack bone and cartilage [33]. Although mice are genetically similar to humans it has been shown that there are critical differences in cellular, molecular and immunological biology [68], thus also questioning the translation relevance of these models for the human disease.

Innovative in vitro 3D models are evolving and becoming translationally relevant, as we understand more about the tumour microenvironment. These 3D models can be constructed from components that are completely human derived, using scaffolds to try and replicate structures of the bone [69]. They are generally cost effective with a high throughput ratio, but the majority do not yet have the complexity to replicate the multicellular interactions attained in OS. The difficulty in creating systems to replicate all human systems, like blood vasculature, is a major limiting factor for in vitro 3D models. Animal and CAM models are able to use the host for processes like angiogenesis and nutrient movement between the tumour and its microenvironment [59]. The continued development of in vitro 3D models may compensate for these issues in the future, but overall these in vitro 3D models currently act as more of a bridge between 2D and in vivo animal models.

Using the CAM model could be an effective method to study OS. This model uses the blood vasculature of the chicken embryo for tumour growth and development [70]. The lymphatic and vascular system is also present in mouse models, but compared to mouse models CAM assays are much more cost effective and easier to use [70,71,72]. Although angiogenesis of the implanted sample is not established by the tissue itself but by the CAM, the formation of blood vessels is very important in tumour growth and development. As well as an inexpensive, easy to use tumour model, the CAM also allows researchers to look at cell-to-cell interactions and metastasis. Current studies of OS are limited to cells in sponges or being implanted directly onto the CAM, by combining OS cell lines with other cells in the microenvironment, most notably structural bone, could give a better insight into these processes and interactions, building on the information obtained by Balke et al. about the generation of OS tumours [59].

All of the non-in vivo models discussed in this literature review are limited by the lack of published data combining OS cells with other cells found in the bone and cartilage niche and tumour microenvironment. Without the interaction of these cells, we cannot effectively mimic the aetiology in a 3D model. There are recent studies that have shown that bone regeneration is possible in the CAM model, Moreno-Jiménez et al. (2018) successfully implanted bone cylinders onto the CAM resulting in bone growth, osteoid deposition, angiogenesis and the generation of mineralised tissue [73]. Current work is focusing on integrating OS cells in a humanised bone ECM model, which can then be implanted onto the CAM for vascularisation (Figure 3). This model could be a robust method for testing activated monocytes, macrophages, osteoclasts and BMSCs in the functional development of OS and could gain a novel insight into effectively replicating the OS tumour microenvironment.

## Figures and Tables

**Figure 1 ijms-21-05499-f001:**
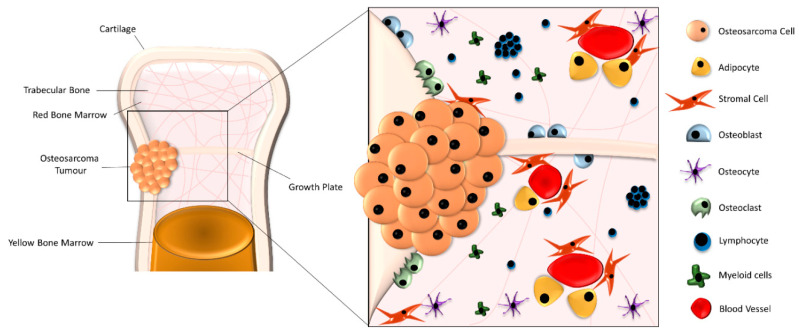
Formation of the osteosarcoma (OS) microenvironment. The OS microenvironment is a complex composition of structural cells, including stromal cells, osteoblasts, osteoclasts, osteocytes and adipose cells, alongside lymphocytes and myeloid immune cells, which are transported through the bone marrow by blood vessels. OS cells routinely form tumours along the growth plate that can protrude from the trabecular bone through the cartilage layer of the bone.

**Figure 2 ijms-21-05499-f002:**
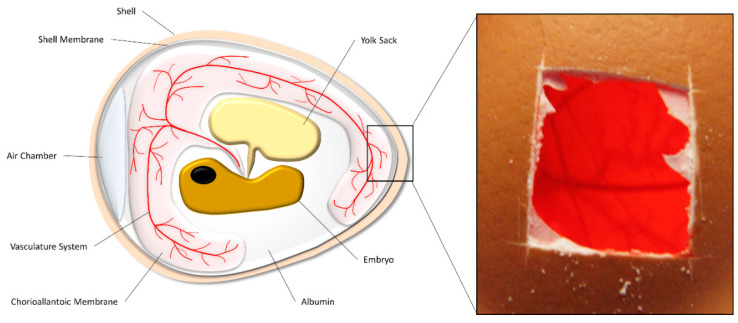
Structural composition of a chicken egg. The chorioallantoic membrane (CAM) is a membrane that covers the inside of the eggshell, where it acts as a respiratory organ for the chick embryo. A window can be made in the shell and the sample of interest placed on the CAM.

**Figure 3 ijms-21-05499-f003:**
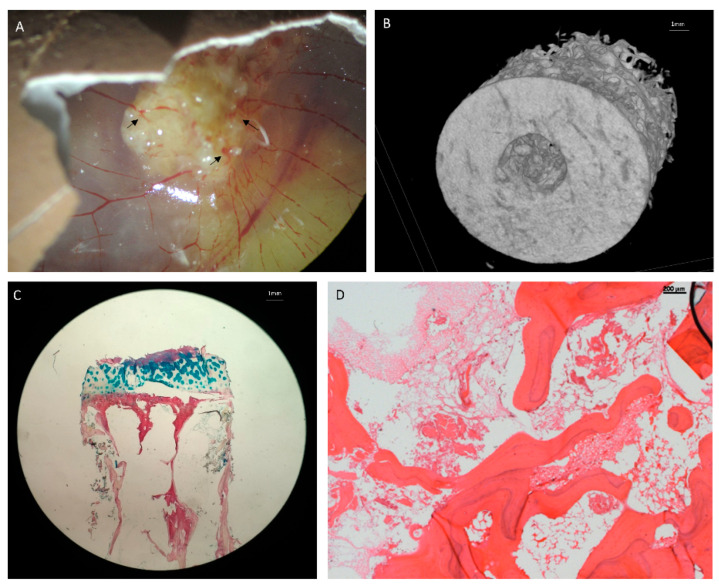
Bone extracellular matrix (ECM) implanted in CAM model for vascularization. (**A**) Human bone inserted into CAM model. Arrows show areas of angiogenesis into the bone. (**B**) A µCT image of the bone model. (**C**) Alcian Blue and Sirius Red staining of the bone ECM, Blue denotes cartilage and proteoglycans, red denotes collagen. (**D**) Hematoxylin and Eosin Y staining of the bone ECM.

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
