# Peer review of "The Role of Pre-Clinical 3-Dimensional Models of Osteosarcoma"

_ijms, 2020, doi:10.3390/ijms21155499_

Round 1
Reviewer 1 Report
In the manuscript “The role of pre-clinical 3-Dimensional models of osteosarcoma” the authors describe the actual models used for the development of osteosarcoma both in vitro and in vivo, showing both the advantages and the limitations of all of them.
The review is well written, practical and concise. It is relevant for researcher and clinicians as well.
Just few minor revisions:
Please levels out the words” in vitro and in vivo “in the text (in the manuscript is always reported “in vitro” while in the conclusion paragraph is written “in vitro)
Phrase 257: modify the sentence “here I suggest”
Reviewer 2 Report
The review provides a comprehensive and well documented revision of the 3-D models developed for osteosarcoma.
Points to be addressed.
_the abbreviated form OS should be used Throughout the manuscript
-The most relevant achievments obtained by the use of the different models should be reported and discussed with particular attention to any potential discrepancies observed.
Round 2
Reviewer 2 Report
The manuscript has been improved adequately and it is now acceptable for publication.